# Exploring Physical Activity Engagement and Related Variables During Pregnancy and Postpartum and the Best Practices for Self-Report Physical Activity Postpartum

**DOI:** 10.3390/ijerph22111711

**Published:** 2025-11-13

**Authors:** Stephanie Turgeon, Iris Lesser, Corliss Bean

**Affiliations:** 1Department de Psychoeducation et de Psychologie, Université du Québec en Outaouais, Gatineau, QC J9A 1L8, Canada; stephanie.turgeon@uqo.ca; 2School of Kinesiology, University of the Fraser Valley, Chilliwack, BC V2R 0N3, Canada; 3Recreation and Leisure Studies, Brock University, St. Catharines, ON L2S 3A1, Canada; cbean@brocku.ca

**Keywords:** motherhood, prenatal period, postnatal period, exercise, active life, children

## Abstract

Physical activity (PA) is recommended in pregnancy and postpartum to support mental and physical well-being. However, little is known about the association between pregnancy and postpartum PA and interrelated factors in PA engagement. The objectives of this study were to (a) measure and understand PA engagement in pregnancy and postpartum and how related variables (i.e., work status, number of children, time since birth, PA during pregnancy) are associated with postpartum PA and (b) to examine two self-reported methods for assessing PA postpartum: self-reported PA volume and intensity through questionnaire vs. asking whether women met PA guidelines of 150 min of moderate-to-vigorous PA per week. A total of 526 women who had given birth within the past 18 months completed an online questionnaire (majority were Canadian or American). Descriptive statistics were used to assess PA during pregnancy and postpartum, and chi-square analyses were run to assess the association between related variables and to evaluate self-report methods. During pregnancy, 27.4% of women reported meeting PA guidelines and 25.3% reported meeting PA guidelines postpartum. No significant relationship between return-to-work status or number of children and meeting PA guidelines was found. Participants ≤12 weeks postpartum were less likely to meet PA guidelines compared to those >12 weeks postpartum. There was a significant relationship between meeting PA guidelines during pregnancy and engagement in PA postpartum. Lastly, there was a significant relationship between a binary measure of meeting PA guidelines (i.e., yes or no) and calculated PA volume and intensity when provided through type, frequency, and duration. This study provides insights into PA patterns of women during pregnancy and postpartum. Findings highlight the need for targeted interventions to support maternal health and well-being, emphasizing the importance of establishing PA habits during pregnancy to assist in maintenance postpartum. Results also suggest that simplified assessment methods may be effective for monitoring women’s PA, potentially making it easier for healthcare providers to track and promote healthy behaviors among new mothers.

## 1. Women Moving Forward: Exploring Physical Activity Engagement During Pregnancy and Postpartum

Physical activity (PA), defined as any bodily movement that increases energy expenditure [1], is associated with a range of positive physical and psychological health outcomes through pregnancy and the postpartum period [2,3,4], as well as the ability to better manage the demands of raising children than those who are inactive [5]. The World Health Organization [6] has created evidence-based PA guidelines to improve health and prevent disease across different age groups and populations [7]. These guidelines emphasize the benefits of being active while also addressing the risks associated with prolonged sedentary behavior. Based on these recommendations, postpartum women should aim to achieve or exceed the PA guidelines of 150 min of moderate-to-vigorous PA per week with a gradual return to PA 4–6 weeks after delivery or with healthcare recommendation [3]. Postpartum is a term which reflects a mother after delivery with recognition that it takes almost one year physiologically to return to the pre-pregnancy stage [8]. Recently, Davenport et al. (2025) created postpartum PA guidelines which emphasize the importance of individualized, gradual, and symptom-based approaches tailored to each person’s readiness for PA [3].

Despite the many noted benefits of PA for women, PA levels are low in pregnancy (mean of 12.3 min per day of moderate-to-vigorous activity) [9] and tend to remain low postpartum [10]. Pregnancy involves more time spent with a healthcare provider which can provide support around PA behavior [11]. PA behavior may be impacted by fatigue, lack of time, pregnancy-related discomforts [12], unclear advice, and concerns about their baby’s health [13]. However, PA is often not discussed with pregnant women unless they are experiencing comorbidities which could be improved through PA [14]. Nevertheless, particularly for first-time mothers, there may be fewer barriers to PA engagement in a healthy pregnancy as one has yet to adjust to their new role as a mother with increased responsibilities, such as sleep deprivation, feeding challenges, and body image concerns [15]. Comparatively, low PA engagement among postpartum women may be due to fatigue, lack of time, low self-efficacy, body dissatisfaction, and/or not being provided adequate evidence-informed information around safely returning to movement [16,17]. Further, postpartum women typically do not maintain contact with a healthcare provider beyond the 6-week check-up [6], reducing opportunities for healthcare provider advice [16]. Examining factors associated with PA postpartum is valuable, given that new mothers may have reduced levels of PA after pregnancy. As time progresses from the birth of a child, postpartum PA may increase, with prospective study designs finding PA returns to pre-pregnancy levels by the 12-month mark [10,18], though walking may remain lower during postpartum than during pregnancy [10]. Reductions in previously established PA from pre-pregnancy through postpartum further suggest that the postpartum period presents new PA challenges [19]; however, some of these challenges may diminish over the postpartum period such as fatigue (which is associated with PA postpartum) [20]. However, new challenges may also arise such as returning to work after having a child which may further impact PA due to added maternal responsibilities [17]. In addition, maternal responsibilities may increase with number of children. Mothers with multiple children have lower levels of moderate-to-vigorous PA than those with one child [21]. This may be exaggerated in the postpartum phase where mothers who have more than one child, including one infant under 6 months of age, were the least likely to meet the PA guidelines compared to mothers of multiple children of older age [22].

Methodology around PA engagement frequently involves assessment of whether one is adhering to the PA guidelines given its simplicity in measurement on a large scale [23]. Given the propensity for the assessment of pregnancy and postpartum PA using this methodology [22,23], it is essential to understand if subjectively assessing whether one is meeting the PA guidelines through a binary variable (i.e., yes or no) coincides with a person’s reported frequency and intensity of PA. Hesketh et al. (2024) noted a discrepancy between absolute measures of PA intensity and perception of intensity during both pregnancy and postpartum [24,25]. Specifically, PA may be misclassified as more vigorous than its true physiological intensity (i.e., associated heart rate and ventilation), as physiological changes in pregnancy make exercise feel harder than its physiological demands and this dissociation may remain up to 12 months postpartum [24]. Given the use of the PA guidelines as a metric of whether pregnant and postpartum women are engaging in adequate PA to achieve health benefits, it is worth examining whether absolute measures of PA intensity are correctly associated with whether women feel that they are meeting PA guidelines at these time points. To date, limited research has explored the role of time since giving birth, return-to-work, and PA engagement during pregnancy on postpartum PA engagement. Therefore, the primary objective of this study was to measure and explore women’s PA behaviors during pregnancy through postpartum and how the PA during pregnancy, return-to-work status, number of children, and postpartum period were associated with women’s PA during the postpartum period. The second objective of this study was to compare two self-reported methods for assessing women’s PA to determine their feasibility for simplistic methods of PA questionnaires in postpartum. Whether women report feeling as though they were meeting PA guidelines across multiple time points in pregnancy and postpartum is unknown.

## 2. Method

### 2.1. Recruitment and Procedure

Institutional ethics approval was received from the University of the Fraser Valley and Brock University Human Research Ethics Board prior to data collection. Participants were recruited through social media and word-of-mouth from June 2021 to January 2022. Participants were eligible for study participation if they identified as (a) a woman, (b) a mother (who had given birth to their child), and (c) were less than 18 months postpartum. Informed consent was acquired electronically through SurveyMonkey. The term “woman” is used to define one person’s gender. Within this study, gender refers to “the socially constructed roles, behaviors, and identities of female, male, and gender-diverse people”. This study was part of a larger survey design examining the role of PA on postpartum physical and psychological well-being. Specific to this study, the questionnaire included (a) socio-demographic information (e.g., age, maternal history, social support, return-to-work) and (b) PA volume as assessed by frequency, duration, and type of weekly activity and whether participants reported meeting the PA guidelines of 150 min of moderate-to-vigorous PA per week.

### 2.2. Measures

#### 2.2.1. Demographic Information

Women were asked a series of questions to provide demographics including age, ethnicity, age of their most recently delivered child, whether they had a vaginal or surgical birth for their last child, number of pregnancies, number of live births, pregnancy risk factors, and any birth complications. Women were also asked whether they were the primary caregiver of their child(ren). Lastly, participants were asked about whether they had returned to work. Note that number of live births was used to calculate number of children (i.e., one live birth = one child, two live births = 2 children).

#### 2.2.2. Physical Activity

Women were asked questions about their PA habits both during pregnancy and postpartum. Specifically, women were asked to describe whether they would describe themselves as physically active, both during pregnancy and postpartum, with the provided definition: ≥150 min of moderate-to-vigorous effort PA such as brisk walking [7]. This is a commonly used PA measure and is used in stratification guidelines for PA [25]. In our more detailed measure of PA, women were asked to provide a thorough overview of their average weekly PA engagement during pregnancy and postpartum. Participants were asked to list the types of PA they engaged in (e.g., walking, strength training, yoga), the frequency of PA (times/week), and the average duration of PA (minutes [min]). To describe the intensity of PA, activity types were classified into metabolic equivalents (MET) as described in the Ainsworth Compendium to estimate the energy cost of PA [26]. Low-intensity PA was classified as <3 METs, moderate-intensity as 3–5.9 METs, and vigorous-intensity (≥6 METs) activities. Using the Ainsworth Compendium PA were categorized as light, moderate, or vigorous to provide a sum of the volume of PA at each intensity. Finally, participants’ moderate and vigorous PA volumes were summed and transformed into a binary score: either meeting PA guidelines (i.e., sum ≥ 150 min/week) or not meeting PA guidelines (i.e., sum < 150 min/week). PA questions were modeled after Davenport’s (2020) [27] methodology, providing a description of the amount and type of PA completed while allowing for computation of the intensity of PA using the Ainsworth Compendium of PA [28]. In addition, participants were asked to answer the following question to quantify their PA behaviors during their current postpartum period: Do you currently meet the recommended amount of PA for postpartum women of 150 min of moderate-to-vigorous PA per week? This question was not included in pregnancy questions.

### 2.3. Analyses

To meet the general objective of the present study, descriptive statistics were used to describe PA during pregnancy and postpartum. To fulfill the secondary analyses, 2 × 2, 2 × 3, and 2 × 2 × 3 chi-square analyses were run using SPSS version 31 (IBM, Armonk, NY, USA). Prior to interpreting the findings, we verified underlining assumptions of the chi-square test. Specifically, the data had to be nominal or ordinal, groups for each variable had to be mutually exclusive, and the theoretical cell counts for all cells had to be ≥5. Effect size for the 2 × 2 analyses was interpreted using the Phi (φ) statistic, while Cramér’s V was used for crosstabs greater than 2 × 2. Interpretation of effect sizes were as follows: [0.10–0.30] = small; [0.30–0.50] = medium; and [0.30–1.00] = large [26]. Chi-square results are presented if all assumptions were met. Missing data (described below) were not input due to the categorical nature of the variables.

## 3. Results

### 3.1. Participants

Our sample included 526 women aged 18 to 45 years (*M* = 28.3). The majority self-identified as White/European (79.8%; *n* = 420), followed by Latin American (6.3%; *n* = 33), with the remaining 13.9% (*n* = 73) representing other ethnic backgrounds. Participants were 0 to 100 weeks postpartum, with 171 (32.6%) classified as “early postpartum” (i.e., 0–3 months), 153 (29.1%) “mid postpartum” (i.e., 3–6 months), and 200 (38.6%) “late postpartum” (i.e., >6 months [18]). Participants reported between zero and six live births (*Md* = 1). Specifically, 55% of women had one child (*n* = 291), 34% had two children (*n* = 179), and 11% had three or more children (*n* = 56). It was found that 60% of participants reported a healthy pregnancy free of clinical diagnoses (25 missing values due to an unanswered question), and 59% were free of complications during the birthing process (12 missing values). The most prevalent clinical diagnoses obtained prior to or during pregnancy were anxiety (*n* = 69), depression (*n* = 52), and gestational diabetes mellitus (*n* = 43). Regarding the complications that occurred during birth, the most frequently reported included fetal distress (*n* = 81) and fetal malposition (*n* = 50). One in four women (*n* = 131; 24.9%) had returned to their full-time occupations (work or university) at the time of data collection. Finally, nearly all women in our sample (*n* = 514; 97.2%) reported being the primary caretaker of their child(ren).

### 3.2. Physical Activity from Pregnancy to Postpartum

#### 3.2.1. Pregnancy

Using PA measures from the PA questionnaire which assessed type, duration, and length of PA sessions, participants reported varying levels of PA. Approximately 85% (*n* = 453) of the participants reported engaging in at least one PA session per week (i.e., they described at least one type of PA with a noted frequency and length). The number of weekly PA sessions varied between zero and 27, with a median of five sessions per week. The total volume of PA was 226.90 min/week (range = 0–1740; *SD* = 236.12). Further dissection of the results showed that participants considered 44.0% (*M* = 98.74 min/week; *SD* = 165.59) of their PA to be low intensity, 40.4% moderate intensity, and 7.5% (*M* = 17.13 min/week; *SD* = 85.63) vigorous intensity; the remaining PA volume was not reported (see Figure 1). Although the mean volume of PA may be high when considering volume intensity, when calculating the sum of participants’ moderate and vigorous PA volumes only 27.4% (*n* = 144) of participants reported that they met the PA guidelines of 150 min of moderate-to-vigorous PA per week during their last pregnancy.

#### 3.2.2. Postpartum

Using PA measures from the PA questionnaire which assessed type, duration, and length of PA sessions, participants also reported their postpartum PA (i.e., following the birth of their last child). A total of 410 participants reported completing at least one PA session per week (*M* = 4.16; *SD* = 3.6; range= 0–20). The mean volume of PA was 172.36 min/week (*SD* = 212.54). When considering volume by intensity, the majority of participants’ PA was considered low (*M* = 72.87 min/week; *SD* = 139.02) or moderate intensity (*M* = 79.63 min/week; *SD* = 174.41). A small proportion of reported PA was considered vigorous (*M* = 19.06 min/week; *SD* = 59.97) (see Figure 1). In the self-report binary measure (yes or no) of whether participants believed they met the postpartum PA guidelines of ≥150 min of moderate-to-vigorous effort PA per week, 218 participants (41.4%) perceived to meet the PA guidelines, while the remaining 308 participants (58.6%) reported that they did not meet the PA guidelines. In parallel, when calculating the sum of participants’ moderate and vigorous PA volumes, 133 (25.3%) participants met the PA guidelines while 391 (74.3%) did not meet the PA guidelines. Two participants had insufficient data to calculate their total volume for moderate-to-vigorous intensity PA. Figure 1 presents total PA volume during pregnancy and postpartum, as well as their PA volume for low, moderate, and vigorous intensity. Continuous data could not be compared due to the large number of participants with volumes equal to zero min/week, thus creating largely skewed distributions for both the observed data and of residuals.

### 3.3. Physical Activity and Associated Variables

There was no significant relationship between participants return-to-work status and meeting the PA guidelines, χ^2^ = 0.41, *p* = 0.524, φ = 0.108. Similarly, participants’ number of children (categorized as 1, 2, 3, or more) was not significantly associated with PA χ^2^ = 0.99, *p* = 0.610, φ = 0.108.

In contrast, a significant relationship was found between participants’ postpartum phase (time since giving birth) and meeting the PA guidelines, χ^2^ = 4.50, *p* = 0.047, Cramér’s V = 0.028. Specifically, participants who were early postpartum (≤12 weeks) were less likely to meet the PA guidelines (24.1%) than participants who were mid postpartum (33.1%; 12–24 weeks) or late postpartum (42.9%; >24 weeks). The strength of the relationship between the postpartum period and participants’ PA engagement was small (Cramér’s V 0.028).

Finally, a significant relationship was found between whether participants met PA guidelines during pregnancy and whether they met PA guidelines during the postpartum period, χ^2^ = 57.96, *p* < 0.001, φ = 0.334. When controlling for participants’ postpartum period using a 3-way chi-square analysis, the relationship between participants’ PA during pregnancy and the postpartum period remained significantly associated at all levels of the postpartum period: early (χ^2^ = 28.20, *p* < 0.001, φ = 0.407), mid (χ^2^ = 18.64, *p* < 0.001, φ = 0.353), and late (χ^2^ = 14.34, *p* < 0.001, φ = 0.269). The strength of these relationships varies from small (φ = [0.10–0.30]) to medium (φ = [0.30–0.50]) effect sizes. A higher proportion of participants who met the PA guidelines during pregnancy also met them during the postpartum period, compared with those who did not meet the guidelines during pregnancy. This relationship was observed across all three postpartum periods. For instance, among women less than 12 weeks postpartum who had met the PA guidelines during pregnancy, 55.6% no longer met the guidelines postpartum, whereas 44.4% continued to meet them postpartum. Conversely, among women less than 12 weeks postpartum who did not meet the guidelines during pregnancy, 91.2% also failed to meet them postpartum, and only 8.8% reported meeting the PA guidelines postpartum (see Table 1).

### 3.4. Reliability of Participants’ Physical Activity Estimation

Two self-reported methods of assessing PA postpartum, the sum of self-reported moderate-to-vigorous PA using a PA questionnaire and whether participants reported meeting the PA guidelines of 150 min of moderate-to-vigorous PA per week (i.e., binary: yes or no), were compared using a chi-square analysis. Our results suggest that the methods for estimating participants’ postpartum PA are significantly related, χ^2^ = 100.57, *p* = < 0.001, φ = 0.438. Specifically, participants who did not meet the guidelines postpartum based on the sum of their reported moderate and vigorous PA volumes were significantly more likely to self-report as not meeting the PA guidelines of 150 min of moderate-to-vigorous PA per week postpartum. As can be seen in Figure 2, the proportion matching results between both assessment methods (e.g., not meeting PA guidelines for both assessment methods: 71.4%) was higher than the proportion of mismatching results between both methods (e.g., not meeting the guidelines based on the PA volume data and self-reporting meeting the guidelines: 28.6%). The strength of the relationship between both methods of assessing participant PA was moderate (φ = 0.438).

## 4. Discussion

This study aimed to examine two objectives. The first was to measure and explore PA behavior during pregnancy through postpartum and how the above variables were associated with women’s PA during the postpartum period. In addition, we assessed the role of time since giving birth, return-to-work, and PA engagement during pregnancy on postpartum PA. The secondary objective was to compare two self-reported methods for quantifying women’s PA to understand whether simplistic PA questions (i.e., binary: yes or no) are appropriate for deducing PA engagement postpartum in comparison to PA volume and intensity reported as a continuous variable. Although many participants reported engaging in at least one PA weekly session, only 27.4% and 25.3% met the recommended PA guidelines (150 min of moderate-to-vigorous PA per week) during pregnancy and postpartum, respectively. These PA levels align with estimates of PA in the United States with 13 to 45% of women meeting PA guidelines during pregnancy [29]. PA engagement, particularly vigorous PA engagement, has been associated with lower postpartum complications than light-intensity PA [30]. As such, it is concerning that only 7.5% of pregnancy PA was classified as vigorous intensity within the current study. As noted by Findley et al., 2020, pregnancy was described as unknown territory when it came to PA practice with women describing limiting PA to reduce any potential harm to their baby [13]. Furthermore, our study and others (e.g., [31]) continue to find lower PA engagement postpartum for women with multiple children than women of a similar age who do not have children [32].

Participants in this study who engaged in PA in their pregnancy were more likely to engage in PA postpartum. This is consistent with previous research (e.g., [33] where women who met the PA guidelines during pregnancy were more likely to meet the guidelines postpartum regardless of time since birth). Given the association between pregnancy and postpartum PA engagement, it is important to advocate for safe PA practices during pregnancy for those without contraindications [34], not only for the mother and infant health benefits [35,36], but also to support PA engagement postpartum. Moreover, results from this study suggest that women in the early postpartum period are less physically active compared to women that are mid-to-late postpartum. This finding aligns with a noted increase in PA from 3 to 12 months postpartum [37]. Lower PA engagement in the early postpartum period may be due to a lack of PA guidance [16] along with other noted barriers to PA engagement such as fatigue, lack of time, low motivation, lack of social support, and body changes [14,38,39,40]. With the recent release of postpartum PA guidelines supporting early return-to-movement (light intensity PA for improving mobilization) [3], there may be a greater adoption of PA behavior during this period. Specific recommendations include beginning or resuming moderate-to-vigorous PA within the first three months, as this has been shown to yield greater benefits for sleep quality and mental health [3].

Finally, returning to work was not significantly associated with postpartum PA levels within our sample. Previously, a small study in Canada found that mothers experienced challenges to PA engagement after returning to work following an extended parental leave [17]. Likewise, a large study sample out of Singapore found an increase in PA levels postpartum due to an uptake in walking through work commuting [10], and therefore, a reduction in time may be accounted for through an increase in lifestyle PA. Neighborhood environment and walkability has previously been found to positively influence PA for pregnancy through postpartum [41]. Variance in PA challenge once a woman returns to work may be due to differences in policy around length of parental leave. Worldwide many mothers return to employment within the first few months after giving birth [42]. We did not find number of children to impact PA engagement in our population. While this does not align with others [21,22], we did not ask the age of other children, and it is largely young children which influence PA engagement [22].

### 4.1. Reliability of Participants’ Physical Activity Estimation

Participants’ perceptions of meeting the postpartum PA guidelines were associated with the values obtained through the sum of their self-reported moderate-to-vigorous PA. However, there is a gap of approximately 15% between reporting methods, with higher percentages when women’s perceptions of meeting PA guidelines are assessed (41.4%) compared to when using their self-reported PA volume (binary: yes or no) (25.3%). Given the need for PA promotion in this population [16] in healthcare settings and the need and desire for greater research in this area [3], assessing varying methodologies in PA measurement postpartum is of importance. To our knowledge there is no validated metric of postpartum PA engagement and this may be a phase of life where greater household and child rearing demands constitute a greater amount of PA that can be considered moderate-intensity PA [35]. In addition, inflation in meeting PA guidelines has been hypothesized to be due to the misclassification of PA as more vigorous during pregnancy and postpartum as physiological changes that may make exercise feel harder than the true metabolic demand up to 12 months postpartum [24]. Similar results have been found in studies conducted with women and mothers with comparable demographic backgrounds as those in our study (e.g., postpartum women; [9]). Given the cost-effectiveness of using self-reported data measures compared to objective PA measures (e.g., accelerometers), future research is needed to identify reliable subjective assessment methods for PA recall. In addition, researchers should consider the limitations associated with the use of binary categorization of PA and self-reported assessment tools.

### 4.2. Study Limitations and Future Directions

Strengths of this study were the large sample size and gathering measures across time points (pregnancy, postpartum), although retrospectively. However, reliance on self-reported PA data comes with certain limitations (e.g., under- or overestimating PA) despite providing insight as to our study objectives. As noted above regarding a paucity of PA questionnaires specific to postpartum, we did not gather specific data related to sedentary behavior and light intensity PA as the nature of the questions refer to structured PA. A large amount of PA postpartum may be household activities (e.g., cleaning, carrying children), in fact, a study completed with 471 women showed that household-related indoor PA and care-giving PA represented 35–70% of their PA from 17 weeks of gestation to 12 months postpartum. As such, it is possible that the number of women who met the 150 min/week PA guidelines are underestimated in our study. Future research using online questionnaires should ensure that questions measure women’s global PA and not only their recreational PA. In addition, nearly 20% of participants reported engaging in 0 min of PA per week, which skewed the distribution of the responses. This situation made it impractical to analyze the data as a continuous variable, thus PA was treated as a binary variable using the World Health Organization (2020) PA guidelines to classify participants [43]. While practical, this approach may oversimplify study findings. For instance, participants completing 100 min of PA/week could have led to significantly different activity patterns and health outcomes compared to those reporting none, yet both participants would fall below the threshold and be included in the same sample. Furthermore, our sample was predominantly White/European (79.8%), which may limit the applicability of the results to more diverse populations [44]. Finally, we did not have any information on women’s PA behaviors prior to their pregnancy. Future research on postpartum PA from pregnancy to postpartum which include preconception PA may provide greater context as to postpartum PA [45]. Lastly, this study took place in late 2021 and early 2022 when COVID-19 restrictions may have still been in place for participants dependent on their geographic location and may have impacted PA behavior or PA behavior in pregnancy when restrictions were more severe [27].

### 4.3. Implications for Practice and/or Policy

Study findings demonstrate that, although insufficient when compared to the PA guidelines, engagement in PA during pregnancy is a significant predictor of sustained PA levels in postpartum. Considering the extensive evidence supporting the physical and psychological benefits of regular PA and highlighting the necessity for timely, targeted guidance and support to aid women in reaching the PA guidelines from early pregnancy to late postpartum [44]. The phase of a woman’s life, from pregnancy through postpartum, has been highlighted as an opportune time for health interventions [46], despite the many barriers to engagement and associated health inequities [44]. Therefore, broad-level support is needed to overcome such barriers to move towards meeting PA guidelines to assist women in achieving long-term health benefits [3]. Encouraging learned PA behavioral strategies through pregnancy such as habit formation, goal setting and planning, and shaping knowledge [47] to support the continuance of postpartum PA are critical. Such effort should be put forward from pregnancy, through late postpartum, and for both women who have returned to work and those who have not. Healthcare providers who work with pregnant and postpartum women should inform and support their patients with establishing PA habits during pregnancy to increase the likelihood of PA engagement postpartum.

In conclusion, our findings suggest that PA levels, particularly at moderate-to-vigorous intensities, remain suboptimal during pregnancy and postpartum. Pregnancy PA shows an association with PA postpartum and PA levels appear to increase over the postpartum period. Lastly, given the consistency between asking women whether they meet the PA guidelines and specific PA questionnaires, this may provide an appropriate starting point for guiding PA in postpartum women.

## Figures and Tables

**Figure 1 ijerph-22-01711-f001:**
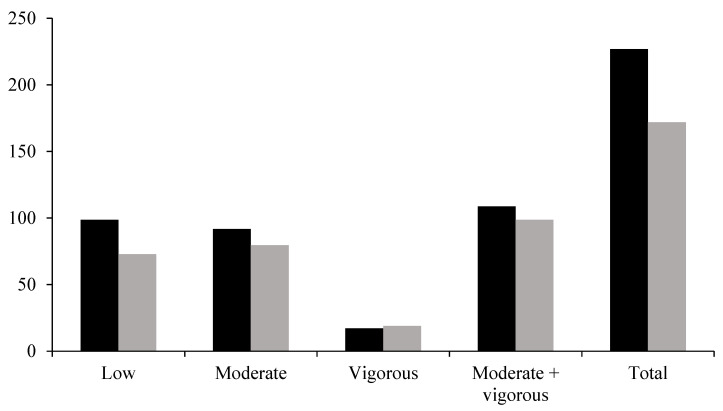
Average self-reported PA volume (minutes per week) during pregnancy and postpartum. Black bars represent pregnancy PA and gray bars represent postpartum PA. The Moderate + vigorous column represents the sum of participants moderate and vigorous PA volumes. The Total column represents the sum of participants low, moderate, and vigorous PA volumes.

**Figure 2 ijerph-22-01711-f002:**
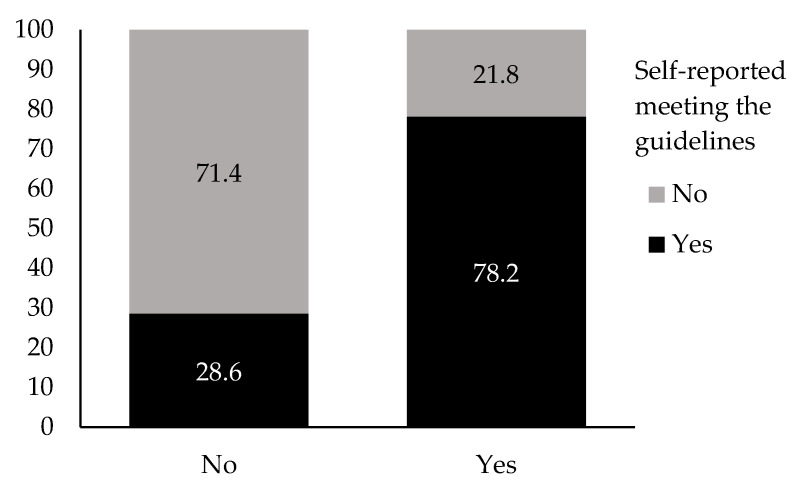
Percent association between meeting or not meeting the PA guidelines of 150 min of moderate-to-vigorous PA per week based on a binary variable of yes or no or the continuous variable depicting self-reported volume and intensity of PA.

**Table 1 ijerph-22-01711-t001:** The table presents the results of the 2 × 2 × 3 chi-square analysis using pregnancy and postpartum PA as a binary variable of meeting or not meeting the PA guidelines. Results are presented by postpartum period (i.e., <12 weeks, 12–24 weeks, >24 weeks) and show the relationship between meeting the PA guidelines during pregnancy with PA postpartum for each of these postpartum periods. The percentage for each line totals 100%, as such the table must be interpreted from left to right. All chi-square analyses are significant at *p* < 0.001.

		Met Guidelines Postpartum	
		No	Yes
Postpartum period	Met guidelines during pregnancy	*n* (%)	*n* (%)
<12 weeks	No	114 (91.2)	11 (8.8)
	Yes	25 (55.6)	20 (44.4)
12–24 weeks	No	89 (80.2)	22 (19.8)
	Yes	17 (43.6)	22 (56.4)
>24 weeks	No	110 (79.7)	28 (20.3)
	Yes	32 (53.3)	28 (46.7)

## Data Availability

The original contributions presented in this study are included in the article. Further inquiries can be directed to the corresponding author.

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
