# Peer review of "Exploring Physical Activity Engagement and Related Variables During Pregnancy and Postpartum and the Best Practices for Self-Report Physical Activity Postpartum"

_ijerph, 2025, doi:10.3390/ijerph22111711_

Round 1
Reviewer 1 Report
Comments and Suggestions for Authors
Thank you for the opportunity to review your article. I enjoyed reading your study and the approach you took. The perinatal period, especially pregnancy, is understudied. It is not surprising that you found that the guidelines were not met during pregnancy. Interestingly, physical activity increased over the postpartum period, but it is not shocking. It is nice to see these trends quantified.
General: The respondents are referred to as women throughout the paper. However, some pregnant individuals do not identify as women. Therefore, please consider using non-gendered terms.
My understanding is that the term Caucasian can be offensive to some groups, and White should be used instead. There is controversy about whether it should be capitalized or not. I would check with the editor of this journal.
At times, I found the results confusing (maybe fatigue played a role). I wonder if more figures or tables would help demonstrate the data.
Figure 1: Would it make more sense to switch the places of vigorous and moderate + vigorous bars?
Line 352: I am unsure why the results are concerning. Do you mean that because the active women continue to be active and the inactive remain inactive, the opportunity to make a lifestyle change was not made during pregnancy?
Author Response
Response to Reviewers
Reviewer 1
Thank you for the opportunity to review your article. I enjoyed reading your study and the approach you took. The perinatal period, especially pregnancy, is understudied. It is not surprising that you found that the guidelines were not met during pregnancy. Interestingly, physical activity increased over the postpartum period, but it is not shocking. It is nice to see these trends quantified.
- Thank you for the positive feedback.
General: The respondents are referred to as women throughout the paper. However, some pregnant individuals do not identify as women. Therefore, please consider using non-gendered terms.
- To be eligible to participate in the study participants had to identify as a woman as noted in the inclusion criterion. Therefore, while we appreciate the feedback, we feel that the term woman is appropriate in the description of participants in this study. See Lines 112-117 for description of the inclusion criteria.
My understanding is that the term Caucasian can be offensive to some groups, and White should be used instead. There is controversy about whether it should be capitalized or not. I would check with the editor of this journal.
- Thank you for this information. As suggested by the American Psychological Association (APA), we have removed Caucasian and replace with “White/European” as in the Results section. As per APA, we have capitalized the “W” and “E”.
At times, I found the results confusing (maybe fatigue played a role). I wonder if more figures or tables would help demonstrate the data.
- Thank you for this suggestion. To make the manuscript more visual we added figures to describe results that were listed in text. Please see Figures 2-5. Each figure presents the distribution of the sample for each of the 2x2 and 2x3 chi-square analyses. Specifically, Figure 2 presents the distribution of the sample for the association between return-to-work status and physical activity guideline adherence postpartum. Figure 3 presents the distribution of the sample for the association between participants’ number of children and physical activity guideline adherence postpartum. Figure 4 presents the distribution of the sample for the association between participants’ postpartum phase and physical activity guideline adherence postpartum. Finally, Figure 5 presents the distribution of the sample for the association between Guidelines Met or Unmet for the Self-Reported and Volume-Intensity Data. For the 2x2x3 chi-square assessing the relationship between participants meeting the PA guidelines during pregnancy and the postpartum period, while controlling for the postpartum period, Table 1 presents the distribution of the sample for this analysis. This table was included in the original manuscript. As such, all analyses in the manuscript are now accompanied by a figure or table.
Figure 1: Would it make more sense to switch the places of vigorous and moderate + vigorous bars?
- Thank you for this suggestion. As we were summing moderate and vigorous PA for the 4th column and then summing the first 4 columns for the last column, we feel it makes more sense to represent the data in its current form.
Line 352: I am unsure why the results are concerning. Do you mean that because the active women continue to be active and the inactive remain inactive, the opportunity to make a lifestyle change was not made during pregnancy?
- Thank you for highlighting this. We agree that the use of the term “concerning” was not appropriate in the context. As such, we have modified the sentences now read as (see Lines 295 to 299): “Although many participants reported engaging in at least one PA weekly session, only 27.4% and 25.3% met the recommended PA guidelines (150 minutes of moderate to vigorous PA per week) during pregnancy and postpartum respectively. These PA levels align with estimates of PA in the United States with 13 to 45% of women meeting PA guidelines during pregnancy (24).”
Reviewer 2 Report
Comments and Suggestions for Authors
This study describes changing physical activity levels during and after pregnancy, and the utility of a simple dichotomous physical activity measure. The relationship between physical activity levels over these time periods is an important area of consideration, however the current study has a number of limitations within their data collection and analysis. Specifically, there is an absence of controlling for important predictor factors (e.g. presence of other children). Furthermore, the introduction and discussion both need further expansion to provide sufficient justification and explanation of the study.
Introduction:
The objectives of the study, in particular the second objective, do not seem to lead logically from the introduction as there is no explanation as to why there is a need for a binary measure of physical activity within the introduction.
Line 44 to 49 needs to add additional information to ensure that the time frame during the perinatal period of these recommendations is clear and as such the relevance to the research question.
Line 58: Needs to provide a definition of postpartum
Results
Reporting of the results is very unclear and needs to be consistent throughout the sections. For example the changing in the way that the activity levels are reported between time periods makes it very difficult for a reader to see where the differences are.
It is also not clear why only returning to work was the only variable assessed where there were other demographic factors, such as the presence of other children, that may have also impacted on physical activity levels. Furthermore, it would have been beneficial to assess return to work by postpartum category given that women who return to work are also likely to be further postpartum.
Line 226-230: Given the changing activity levels with time postpartum, it would be valuable to see the activity levels by each postpartum category.
Discussion
The discussion does not address the many other factors that are likely to be related to PA engagement in early postpartum period. The discussion of low PA engagement in the early postpartum period as due to lack of PA guidance with no consideration of other factors e.g. adjusting to a newborn, physical healing, fatigue (Line 287) highlights this deficiency.
The relationship between postpartum/pregnancy physical activity and pre-existing physical activity needs to be acknowledged within the discussion.
Study limitations
The major limitation present in the study in that physical activity levels were not measured prior to the study needs to be acknowledged and its implications discussed.
The study took place while COVID-19 restrictions were in place. The impact of this upon physical activity levels needs to be acknowledged- ideally this would include indicating what the restrictions were that were in place and the potential impacts of this upon physical activity, including any evidence from your region.
Comments on the Quality of English LanguageThere are a number of language errors throughout the manuscript that impede the ability to understand the manuscript.
Abstract
Line 22: Influence is not the correct term to describe a correlation analysis. It would be better to refer to this as describe.
Line 53 & Line 61-62: its not clear what this sentence means
Line 64-66: What is the comparator for which ‘fewer’ is based upon, i.e. who is it compared to
Line 91: the use of the word warranted makes the sentence unclear
Line 100: It is unclear what ‘the above variables’ refer to precisely
Author Response
Reviewer 2
This study describes changing physical activity levels during and after pregnancy, and the utility of a simple dichotomous physical activity measure. The relationship between physical activity levels over these time periods is an important area of consideration, however the current study has a number of limitations within their data collection and analysis. Specifically, there is an absence of controlling for important predictor factors (e.g. presence of other children).
- Thank you for this feedback. We would like to note that the variables included in our analyses were either binary or ordinal variables, and therefore, we are limited in using control variables in our analyses. However, we agree that the number of children may be a confounding factor and therefore have added this into our analyses. However, we did not include “postpartum period” when running a chi-square between “return-to-work” and “postpartum physical activity” as these variables were significantly related and most likely products of time. Further explanations are presented in the Results section below.
Furthermore, the introduction and discussion both need further expansion to provide sufficient justification and explanation of the study.
- We have addressed each of these concerns and have added further empirical support of the study in the Introduction and Discussion. All changes to these sections are highlighted in yellow. For example, in the introduction, we have specified that
“. Pregnancy involves more time spent with a healthcare provider which can provide support around PA behavior (11) which is impacted by fatigue, lack of time, and pregnancy-related discomforts (12), unclear advice, and concerns about their baby’s health (13). However, PA is often not discussed with pregnant women unless they are experiencing comorbidities which would be improved through PA engagement (14).” (see lines 60-64). We have further justification for the study in the discussion: e.g.: “Lower PA engagement in the early postpartum period may be due to a lack of PA guidance (16) along with other noted barriers to PA engagement such as fatigue, lack of time, low motivation, lack of social support and body changes (14,37–39)” (see line 316-319).
- Moreover, we have added literature on the inclusion of number of children in the analysis (see lines 85-89): “In addition, maternal responsibilities may increase with number of children. Mothers with multiple children have lower levels of moderate-to-vigorous physical activity than those with one child (20). This may be exaggerated in the postpartum phase where mothers who have more than one child, including one infant under 6 months of age, were the least likely to meet the PA guidelines compared to mothers of multiple children who are older (21). “
Introduction:
The objectives of the study, particularly the second objective, do not seem to lead logically from the introduction as there is no explanation as to why there is a need for a binary measure of physical activity within the introduction.
- Thank you for this feedback. We agree that this could be enhanced. Further explanation of this secondary objective is provided with the following addition (see lines 90-95): “Methodology around PA engagement frequently involves assessment of whether one is adhering to the PA guidelines given its simplicity in measurement on a large scale (23). Given the propensity for the assessment of pregnancy and postpartum PA using this methodology (22,23) it is essential to understand if subjectively assessing whether one is meeting the PA guidelines through a binary variable (i.e., yes or no) coincides with a person’s reported frequency and intensity of PA.”
Line 44 to 49 needs to add additional information to ensure that the time frame during the perinatal period of these recommendations is clear and as such the relevance to the research question.
- Thank you. We believe the below addition reflects a time frame which addresses your point (see lines 52-54): “Postpartum is a term which reflects a mother after delivery with recognition that it takes almost one year physiologically to return to the pre-pregnancy stage (6).”
- Additionally, a statement has been added to implications for practice and policy to reflect the need for a defined postpartum timeline (see lines 402-405): “Furthermore, consideration of the timeline postpartum in which additional support is needed should be an area of additional research with definitions of postpartum deviating globally (6).”
Line 58: Needs to provide a definition of postpartum
- Thank you. The following has been added to provide a current definition of postpartum (lines 52-54): Postpartum is a term which reflects a mother after delivery with recognition that it takes almost one year physiologically to return to the pre-pregnancy stage (6).
Results
Reporting of the results is very unclear and needs to be consistent throughout the sections. For example the changing in the way that the activity levels are reported between time periods makes it very difficult for a reader to see where the differences are.
- Thank you for this comment, we have ensured that the results are clearly presented. As suggested by reviewer 1, we have added figures to support the interpretation of the results. We have also added some information in the methods when describing the physical activity measures to further clarify potential areas of confusion. Finally, we have removed a paragraph in the Physical Activity from Pregnancy to Postpartum section that was redundant with results presented in the Reliability of Participants Physical Activity Estimation. The information removed ensures that the results for both the postpartum and pregnancy periods in the Physical Activity from Pregnancy to Postpartum section are more coherently presented.
It is also not clear why only returning to work was the only variable assessed where there were other demographic factors, such as the presence of other children, that may have also impacted on physical activity levels. Furthermore, it would have been beneficial to assess return to work by postpartum category given that women who return to work are also likely to be further postpartum.
- We have added the analysis pertaining to number of children as requested and included the rationale in the introduction and description of the variable in the discussion. We did not, however, include the analysis between return-to-work status and postpartum period as there was significant relationship between the post-partum period and participants return-to-work status. Specifically, those who had returned to work were further in their post-partum period. Both variables seem to be influenced by “time”; thus, including these two related variables in a 2x2x3 chi-square analysis may lead to misleading results.
Line 226-230: Given the changing activity levels with time postpartum, it would be valuable to see the activity levels by each postpartum category.
- Thank you for this comment. The comment aligns perfectly with reviewer 1’s comment to add figures or tables for the different analyses presented. We have added bar charts have been added for each of our analyses (see Figure 2-5).
Discussion
The discussion does not address the many other factors that are likely to be related to PA engagement in early postpartum period. The discussion of low PA engagement in the early postpartum period as due to lack of PA guidance with no consideration of other factors e.g. adjusting to a newborn, physical healing, fatigue (Line 287) highlights this deficiency.
- Given the above addition regarding number of children we have added the following to the discussion (Lines 320-322): We did not find number of children to impact PA engagement in our population. While this does not align with others (21,22), we did not ask the age of other children, and it is largely young children which influence PA engagement (22).
- In addition, we have added the following sentence and associated references to expand on the challenges with PA postpartum (Lines 303-306): Lower PA engagement in the early postpartum period may be due to a lack of PA guidance (16) along with other noted barriers to PA engagement such as fatigue, lack of time, low motivation, lack of social support and body changes (14,37–39).
The relationship between postpartum/pregnancy physical activity and pre-existing physical activity needs to be acknowledged within the discussion.
- As suggested, we have added this to our limitations and future directions. See below for language.
Study limitations
The major limitation present in the study in that physical activity levels were not measured prior to the study needs to be acknowledged and its implications discussed.
- As suggested, we have added this issue into our limitations and future directions (see lines 358-361): “Furthermore, our sample was predominantly White/European (79.8%), which may limit the applicability of the results to more diverse populations (36). Finally, we did not have any measurements on women’ PA behaviors prior to their pregnancy. Future research on postpartum PA from pregnancy to postpartum which include preconception PA may provide greater context as to postpartum PA (37).”
The study took place while COVID-19 restrictions were in place. The impact of this upon physical activity levels needs to be acknowledged- ideally this would include indicating what the restrictions were that were in place and the potential impacts of this upon physical activity, including any evidence from your region.
- Thank you for this comment. You are correct that this may have influenced responses if participants were in a location where restrictions were still present. As we cannot comment on participant location and specific restrictions, we have added this as a limitation of the study. See below addition (Lines 362-365): “Lastly, this study took place in late 2021 and early 2022 when COVID-19 restrictions may have still been in place for participants dependent on their geographic location, which may have impacted PA behavior within or beyond pregnancy when restrictions were more severe (38).”
Round 2
Reviewer 2 Report
Comments and Suggestions for Authors
Introduction: Line 70 information should be replaced with advice
Methods: To answer the objectives of the paper, and in light of the information in results line 226 to 228 an explanation should be provided as to why chi square analyses was used as opposed to a regression or other modelling.
Methods: In relation, to the relationship to the answer regarding return to work status and post-partum period, how did the authors correct for the potential confounding of return to work status with postpartum period. We did not, however, include the analysis between return-to-work status and postpartum period as there was significant relationship between the post-partum period and participants return-to-work status. Specifically, those who had returned to work were further in their post-partum period.
Figure 2 and Figure 5 are difficult to understand. Is there a better way of presenting this data.
Author Response
Thank you for taking the time to provide a secondary review of this manuscript. We have gone through the manuscript in detail to ensure that the quality of the language is improved and that there is consistency in terminology. We have also added better descriptions of the results and captions of figures and tables to improve understanding of the results. Specific responses to reviewer comments are noted below:
Methods: To answer the objectives of the paper, and in light of the information in results line 226 to 228 an explanation should be provided as to why chi square analyses was used as opposed to a regression or other modelling.
Response: Thank you for this comment. As noted on line 222-224: Continuous data could not be compared due to the large number of participants with volumes equal to zero min/week; thus, creating largely skewed distributions for both observed data and of residuals. Due to this we used chi square analyses as this would be the appropriate statistical method for two categorical variables.
Methods: In relation, to the relationship to the answer regarding return to work status and post-partum period, how did the authors correct for the potential confounding of return to work status with postpartum period. We did not, however, include the analysis between return-to-work status and postpartum period as there was significant relationship between the post-partum period and participants return-to-work status. Specifically, those who had returned to work were further in their post-partum period.
Response: Thank you for this follow up. We were unable to control for return-to-work status as we are limited in performing Chi square analyses in the number of control variables that can be included (this is contrary to regression analysis or an analysis of variance).
Figure 2 and Figure 5 are difficult to understand. Is there a better way of presenting this data.
Response:
Figure 1 has been updated to remove axis labels and legend with a better descriptor in the figure caption and the figure added within the result text. The caption now reads as:
“Average self-reported PA volume (minutes per week) during pregnancy and postpartum. Black bars represent pregnancy PA and grey bars represent postpartum PA. The Moderate + vigorous column represent the sum of participants moderate and vigorous PA volume. The Total column represents the sum of participants low, moderate and vigorous PA volume.”
Figure 2 has been removed as this was a non significant finding that is described in text and not needed visually.
Figure 3 has been removed as this was a non significant finding that is described in text and not needed visually.
Figure 4 has been removed and the data is now added into the results section.
Table 1 has been improved in its description to better guide readers. The following has been added to the results section:
“A higher proportion of participants who met the PA guidelines during pregnancy also met them during the postpartum period, compared with those who did not meet the guidelines during pregnancy. This relationship was observed across all three postpartum periods. For instance, among women less than 12 weeks postpartum who had met the PA guidelines during pregnancy, 55.6% no longer met the guidelines postpartum, whereas 44.4% continued to meet them. Conversely, among women less than 12 weeks postpartum who did not meet the guidelines during pregnancy, 91.2% also failed to meet them postpartum, and only 8.8% reported meeting the PA guideline (See Table 1).”
The following has been added in caption below the table:
Table 1. The table present the results of the 2 x 2 x 3 chi-square analysis using pregnancy and postpartum PA as a dichotomous variable of meeting or not meeting the PA guidelines. Results are presented by post-partum period (i.e., <12 weeks, 12-24 weeks, >24 weeks) and show the relationship between meeting the PA guidelines during pregnancy with PA postpartum for each of these postpartum periods. The percentage for each line totals 100%, as such the table must be interpreted from left to right. All chi-square analyses are significant at p < .001.
Figure 5 is now Figure 2 with improvements to the text below it. Below has been added to the results section to better describe the figure.
“Specifically, participants who did not meet the guidelines postpartum based on the sum of their reported moderate and vigorous PA volumes were significantly more likely to self-report as not meeting the PA guidelines of 150 min of moderate-to-vigorous PA per week postpartum. As can be seen in Figure 2, the proportion matching results between both assessment methods (e.g., not meeting PA guidelines for both assessment methods; 71.4%) was higher than the proportion of mismatching results between both methods (e.g., not meeting the guidelines based on the PA volume data and self-reporting meeting the guidelines; 28.6%).”